# Social familiarity strengthens neural and vocal responses to conspecific calls in zebra finches

Carlos M. Gomez-Guzman, Daniela Vallentin ◉‡*, Jonathan I. Benichov ◉¤‡

Neural Circuits for Vocal Communication Research Group, Max Planck Institute for Biological Intelligence, Seewiesen, Germany

‡ These authors are co-senior authors on this work.
¤ Current address: Institute of Biology Leiden, Leiden University, Leiden, The Netherlands
* daniela.vallentin@bi.mpg.de

## Abstract

Across animals, dyadic vocal interactions often occur within complex acoustic environments containing numerous signalers. The influence of socially relevant acoustic signals on the neural circuits controlling interactive vocal behavior remains poorly understood. We examined this issue in zebra finches, highly social songbirds that maintain nearly continuous vocal contact through the exchange of short innate calls. We developed a behavioral paradigm that elicits differential responses to familiar and unfamiliar vocal partners, enabling the prediction of social context based on individual birds' response patterns. We then used high-density Neuropixels probes to record neural activity within a vocal premotor nucleus in the songbird forebrain, while birds listened to familiar and unfamiliar contact calls. We found that the activity of putative projection neurons and interneurons in this vocal premotor nucleus was modulated by the familiarity of heard calls, with interneurons exhibiting stronger responses to familiar calls. Furthermore, we found that measures of vocal responsiveness correlated with neural response parameters during listening. Specifically, we observed that increased vocal response rates, rapidity, and temporal consistency for familiar call playbacks were correlated with elevated mean and peak firing rates, as well as prolonged activity, in HVC interneurons. These results demonstrate how socially salient auditory information can affect a forebrain premotor circuit to maintain the specificity of vocal interactions within complex and dynamic social environments.

## Author summary

Vocal interactions frequently occur in social groups where the ability to quickly identify and respond to specific individuals is crucial. Zebra finches, known for their highly social nature and vocal behavior, serve as an excellent model for exploring this process. In this study, we investigated how a vocal control region in the bird's forebrain is affected by the vocalizations of a familiar partner versus an

**Data availability statement:** Data are available here: https://github.com/vallentinlab/SocialContext_calls_HVC.

**Funding:** This study was supported by DFG Research Unit 5768: VA 742/7-1 – 532521431 (to D.V.); DFG Research Consortium SFB 1315: 327654276 (to D.V.); DFG Research Grant: BE 7545/1-1 – 459524793 (to J.I.B.); DFG Research Grant: VA 742/6-1 – 547921981 (to D.V.); and the European Research Council Starting Grant: (ERC)-2017-StG-757459 MIDNIGHT (to D.V.). The funders had no role in study design, data collection and analysis, decision to publish, or preparation of the manuscript.

**Competing interests:** The authors have declared that no competing interests exist.

unfamiliar bird, and how this distinction influences the bird's vocal responses. We designed a behavioral test in which zebra finches respond differently to familiar versus unfamiliar call playbacks. The birds responded faster, more often, and with greater timing precision to calls from known partners. To uncover the neural correlate of this effect, we recorded the activity of a premotor brain area called HVC as birds listened to playbacks of familiar and unfamiliar calls. We discovered that HVC neurons, particularly the inhibitory interneurons, showed stronger and more sustained responses to familiar calls. Furthermore, the intensity of these neural responses was a reliable predictor of how quickly and consistently the birds replied in the behavioral test. Our findings reveal how socially meaningful sounds can shape premotor activity to guide context-specific communication within dynamic social groups.

## Introduction

Animal communication plays a critical role in maintaining social cohesion [1,2], coordinating behaviors [3], and signaling relevant social information, such as identity or behavioral state [4]. In particular, reciprocal vocal interactions, or vocal turn-taking, requires finely tuned acoustic processing and accompanying controlled production of context-dependent vocal replies [5–9]. Zebra finches, for instance, produce contact calls to engage in coordinated interactions with other conspecifics [10,11]. These contact calls are used to maintain pair bonding [12,13], and zebra finches can distinguish between individuals by their unique vocalizations [4,14,15]. During turn-based interactions, birds vocalize more rapidly, more frequently, and in a more coordinated manner when responding to familiar individuals [16–19]. However, the neural mechanisms that link the auditory processing of individual calls with subsequent adjustments in vocal response times, remain unknown.

In zebra finches, the forebrain vocal motor pathway, specifically inhibitory interneurons in the cortical vocal premotor nucleus HVC (proper name), regulate the timing of call interactions by modulating premotor activity and thereby influencing the initiation of vocalizations in response to a vocal partner [8,20,21]. In addition to inhibitory interneurons, HVC contains three populations of excitatory projection neurons, one projecting to Area X ($HVC_X$), a striatal nucleus involved in song learning [22], a second population ($HVC_{RA}$), targeting the robust nucleus of the arcopallium (RA), the output nucleus of the forebrain song production pathway [23], and a third one ($HVC_{Av}$) projecting to the auditory area Avalanche [24]. In the context of vocal interaction, $HVC_{RA}$ neurons and downstream RA neurons burst prior to vocal motor output [8,14], while interneurons that inhibit this premotor activity appear to regulate whether and when a call is initiated [8,21]. $HVC_X$ and $HVC_{AV}$ neuron activity is less well explored in the context of vocal interactions. Pair-specific vocal turn-taking involves not only precise control of call timing, but also requires context-dependent responsiveness [25], influenced by factors such as caller familiarity [18]. However, it remains unclear whether or how the individual-specific properties of heard calls might be reflected in the activity of HVC neurons to modulate vocal responses to different callers.

Although specialized for precise vocal motor control of courtship song production, HVC neurons also integrate sensory information [26] through afferents carrying auditory information from the nucleus interfacialis (Nif) [27]. Various studies have documented auditory-evoked activity in HVC neurons in response to call stimuli [21,28,29]. And, while Ma, et al. [28] propose that call-evoked patterns in HVC represent a predictive signal that helps in coordinating communication, the specific information encoded by this activity has yet to be characterized. Call-evoked activity in HVC, a region primarily involved in vocal motor production, might therefore provide a window to understand how socially relevant auditory information guides specific vocal responses in an interactive context.

Here, we developed a behavioral paradigm to bias zebra finch responses to familiar versus unfamiliar vocal partners. After eliciting this vocal response specificity, we were able to investigate its neural basis, specifically by asking whether HVC neurons also exhibit distinct activity patterns depending on the heard caller's identity. We performed high-density extracellular recordings in awake birds and found that HVC interneurons and projections neurons showed auditory-evoked activity, which was differentially modulated by stimulus familiarity. Interneurons were more strongly and consistently engaged by familiar calls. These findings suggest that HVC interneurons play a role in relaying information about the social relevance of other vocalizing individuals, providing a mechanistic bridge between the sensory and motor circuits enabling context-dependent vocal communication.

## Results

### Specificity of vocal replies

Zebra finches frequently exchange short innate calls (Fig 1a) [17,30,31], which can convey information about individual identity [4], and elicit individual-specific call responses [11,12,14]. To examine how this is achieved, we developed a behavioral paradigm to bias birds' vocal responses to the calls of specific individuals. First, we co-housed opposite-sex pairs of zebra finches and recorded their harmonic "stack" calls during vocal interactions. After at least five days of co-housing, the male zebra finches were individually presented with pseudorandomized playback sequences of "stack" calls, including those of their familiar female cage-mate as well as unfamiliar conspecifics (Fig 1b, see Methods). Call responses were recorded during 4 days of playback sessions. While the overall frequency of vocal responses declined over successive days (S1a Fig), we observed distinct vocal patterns in response to familiar vs. unfamiliar calls (Fig 1c), differences that persisted throughout the experimental days (S1 Fig). We found that response probabilities were higher when birds heard familiar call playbacks, compared to unfamiliar calls (Fig 1d). Moreover, response latencies to familiar call playbacks were shorter (Fig 1e) and exhibited less temporal variability (standard deviation, Fig 1f), replicating previous findings that zebra finches respond more reliably to affiliated partners [16–18]. When comparing how birds responded to unfamiliar calls based on the sex of the caller, we observed distinct patterns. Birds reacted differently to unfamiliar female calls than they did to unfamiliar male calls. Specifically, when responding to female calls, the response latency changed significantly between familiar and unfamiliar females (S2 Fig). This suggests that males adjust particular aspects of their vocal responses when interacting with females [17]. To test if the measured vocal response parameters systematically encode information about the familiarity of heard calls, we trained a random forest classifier to categorize playback familiarity using the observed response patterns above. The resulting classifier could accurately predict whether a particular call playback was familiar or unfamiliar based on a given bird's responses (accuracy = 79.71 ± 11.32%, Fig 1g), indicating that the zebra finches' vocalizations reflect social relevance rather than the inherent features of the acoustic signals, alone (S3 Fig). Together, these results suggest that social experience can influence the perceived salience of a given call and the specificity of birds' vocal responses.

### Neural activity in vocal premotor nucleus HVC is differentially modulated by caller familiarity

We next investigated whether neural activity in the vocal premotor nucleus HVC is affected by the familiarity of heard calls. We conducted high-density Neuropixels probes recordings in HVC of awake head-fixed male zebra finches while

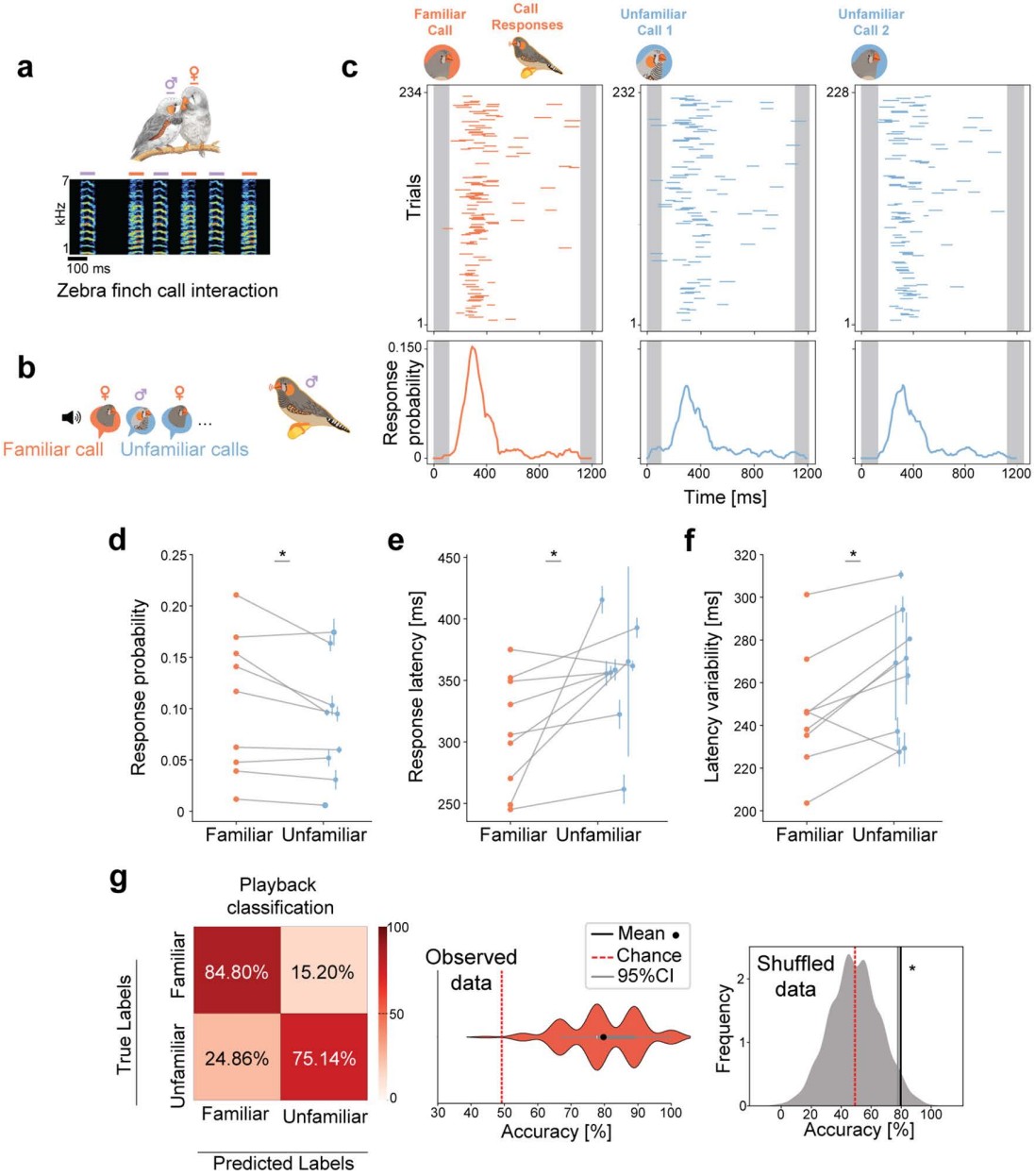

**Fig 1. Vocal behavior is modulated by the familiarity of specific callers. (a)** Example call interaction between a bonded pair of zebra finches in a colony. **(b)** Experimental setup depicting a male zebra finch presented with playbacks of calls from a familiar and unfamiliar conspecifics in a pseudo-randomized manner. **(c)** Top panels: Call responses (colored lines) to call playbacks (grey intervals) presented once per second (exemplified by data from the initial two days). Bottom panels: Call response distributions across inter-playback intervals (calculated over four days/sessions). **(d)** Peak response probabilities (max response probability in response time distribution) to different playbacks (n = 9, birds = 7, days = 4). Response probability (familiar)=0.117, response probability (unfamiliar)=0.090, Wilcoxon signed-rank test, p = 0.03. **(e, f)** Response latencies (from playback onset to response onset) and response latency variability (std of latencies to all responses). Response latency (familiar)=306ms, response latency (unfamiliar)=354ms, Wilcoxon signed-rank test, p = 0.01. Latency variability (familiar)=246ms, latency variability (unfamiliar)=264ms, Wilcoxon signed-rank test, p = 0.02. Blue solid dots represent mean values across multiple unfamiliar playbacks, and error bars standard error (sem). **(g)** Average classification accuracy for playback familiarity based on behavioral features described in d, e and f (model = random forest, iterations = 1000, test size = 0.5). Left: Confusion matrix. Middle: Distribution of accuracies across runs (79.71 ± 11.32%). Right: Kernel density estimate distribution derived from shuffled data. The solid gray line indicates the 95% confidence interval of the shuffled distribution (77.78%), while the black solid line represents the mean accuracy of the observed data (79.71%, p = 0.016). Chance level = 49.2%. * denotes p < 0.05.

pseudo-randomly presenting call playbacks from familiar and unfamiliar conspecifics. From all recordings performed (n = 9 sessions, birds = 8), we acquired activity from 210 putative interneurons with narrow spike waveforms (firing rate (interneurons)=9.51 ± 8.37Hz), and 555 broad spike waveforms from putative projection neurons (firing rate (projection neurons)=1.93 ± 3.8Hz, S4 Fig, see Methods). We found that 80.47% (169/210) of the recorded interneurons (Fig 2a) and 72.07% (400/555) projection neurons (Fig 3a) exhibited call playback-evoked changes in activity, with firing rates increasing during playback presentations, and often remaining elevated after stimulus offsets. In addition, the majority of neurons with call-related activity responded to both familiar and unfamiliar stimuli. However, when comparing response proportions, around 3.5% more neurons showed biases towards familiar sounds. For interneurons, 88.75% (150/169) responded to familiar playbacks, and 85.2% (144/169) to unfamiliar playbacks (Fig 2b). For projection neurons, 83% (332/400) responded to familiar playbacks, and 79.25% (317/400) to unfamiliar (Fig 3b). As controls, we presented a 20 kHz pure tone pulse and a "silent playback", neither of which induced changes in firing rate in any recorded neuron (S5 and S6 Figs).

To examine how neural activity evolves over time in response to familiar and unfamiliar calls, we analyzed the firing rates within the 400ms following playback onsets. Average neuron activity showed an initial transient peak shortly after playback onset (until approx. 50ms), followed by a lower and slowly decaying sustained elevation in firing rate persisting > 100ms after playback offsets (Figs 2c and 3c). To assess whether distinct response patterns emerged for different call types, we visualized the temporal trajectories of responsive neurons in a dimensionally reduced space (PCA, see methods). For both neuron types, the overall response dynamics for familiar stimuli extended across a larger area within each PC space (Figs 2d left and 3d left), suggesting that neural activity differs with caller familiarity. To identify key time points at which activity differed the most, we calculated the Euclidean distances between response trajectories, over time. Both, interneuron and projection neuron trajectories showed sustained differences across conditions, from playback onset until ~350ms after playback offsets (Figs 2d right and 3d right). These findings highlight that neural activity differentiating familiar and unfamiliar calls emerges rapidly, but can remain past stimulus offsets, suggesting a persistent representation of caller-specific information.

We then investigated whether the observed differences in neural dynamics contained sufficient information to decode the familiarity of the triggering call playback. We trained a support vector machine classifier and predicted call playback familiarity with an accuracy higher than chance only if based on the activity patterns of interneurons (interneuron accuracy: 61.1 ± 7.78%, Fig 2e, projection neuron accuracy: 53.85 ± 5.07%, Fig 3e), pointing to a stronger modulation of interneuron activity by caller familiarity, compared to projection neurons. To test whether the classifier solely relies on the weight of the longer response latency of the interneurons, we split the neural activity into two distinct periods 1) during playback (0–100ms) and 2) after playback (100–400ms). Using either of these timeframes individually resulted in worse classification accuracy compared to using the entire dataset (S7 Fig). When including only the late responses, the classifier was unable to reliably predict which call was presented, indicating that the classifiers accuracy in Fig 2e does not exclusively rely on the contribution from long responding interneurons.

Next, to understand what neural activity parameters drove the observed differential modulation, we compared the mean firing rate, maximum firing rate, time of maximum firing rate and the response duration for all neurons during each condition (time window: 400ms following playback onsets). Using a linear mixed model to account for variability across neurons and birds, we found that interneurons exhibited significantly higher mean firing rates, and longer periods of modulated activity in response to familiar call playbacks (Figs 2f and S8), suggesting that interneurons are more strongly and consistently engaged by familiar calls. However, the peak firing rate and the time at which activity reached its maximum did not differ between conditions (Figs 2f and S8), indicating that while the overall intensity and duration of interneuron responses are modulated by familiarity, the latencies with which auditory information is processed by interneurons remain constant. Projection neurons only differed in response duration across conditions (Figs 3f and S8), thus appeared to encode less information about familiarity than interneurons (Figs 2e, 3e, and S9).

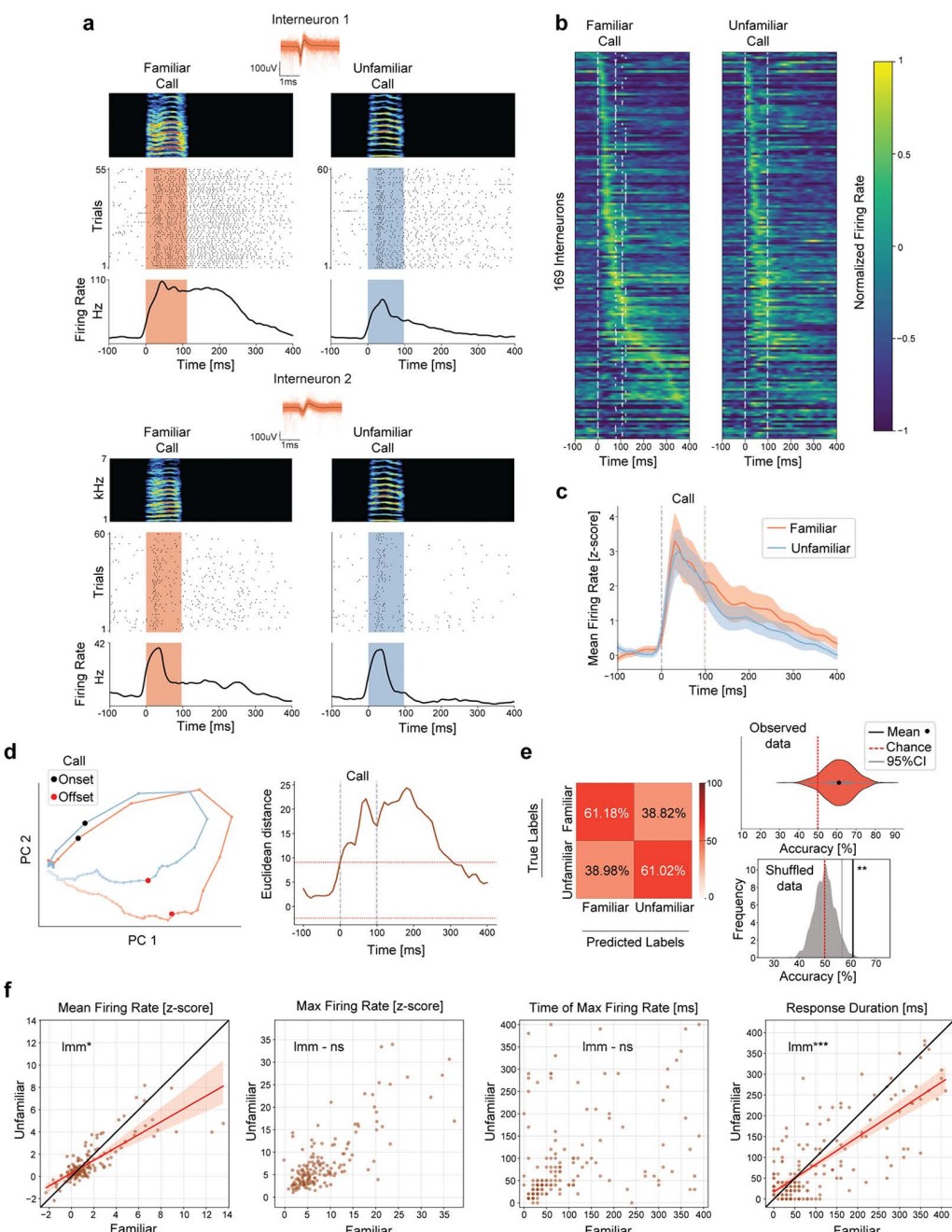

**Fig 2. Interneuron activity is differentially modulated by caller familiarity. (a)** Example cells recorded during familiar and unfamiliar call playbacks. Top: Spike-sorted waveforms with average (black) and spectrogram of call playback. Middle: Spike dot raster plot. Bottom: peri-stimulus time histogram (PSTH). **(b)** Normalized firing rate for interneurons that changed activity beyond 2 standard deviations from baseline (169/210 Interneurons, recordings = 9, birds = 8). Neurons are ordered by their peak firing time during familiar call playbacks. The same neuron order is maintained for unfamiliar call playback, showing corresponding activity patterns across conditions. White dashed lines depict call onsets and offsets. **(c)** Average normalized firing rate (z-score) across significantly responsive neurons. Shaded area represents the 95% confidence intervals (1.96*standard error). **(d)** Left: Response trajectories in PC space for neurons shown in b (Variance explained by first 2PCs = 55.34%). Lines connecting dots represent 10ms. Right: Euclidean distance between conditions in PC space across time. Red dotted lines represent ±2std from mean baseline values. **(e)** Average classification accuracy for call playback familiarity based on the firing rate of neurons shown in b (time window = 0 to 400ms from playback onset, model = support vector machine, iterations = 1000, test size = 0.1). Left: Confusion matrix. Top Right: Distribution of accuracies across runs (61.1 ± 7.78%). Bottom Right: Kernel density estimate distribution derived from shuffled data. The solid gray line indicates the 95% confidence interval of the shuffled distribution (56.78%), while the

black solid line represents the mean accuracy of the observed data (61.1%, permutation test, p = 0.005). Chance level = 50%. **(f)** Distribution of different features extracted from the neural activity. Conditions compared using linear mixed model (lmm, see S8 Fig). Firing rate (familiar) = 1.46 ± 2.32 (z score), firing rate (unfamiliar) = 1.14 ± 1.65 (z score), lmm, p = 0.011. Max firing rate (familiar) = 8.4 ± 6.9 (z score), max firing rate (unfamiliar) = 7.63 ± 6.38 (z score), lmm, p = 0.051. Time of max firing rate (familiar) = 94.85 ± 103.51 ms, time of max firing rate (unfamiliar) = 88.52 ± 96.22 ms, lmm, p = 0.491. Response duration (familiar) = 108.87 ± 112.98 ms, response duration (unfamiliar) = 88.87 ± 95.62 ms, lmm, p = 0.001. Black line depicts the identity line where slope = 1. Red line represents fitted regression line for significant comparisons and shaded region shows the 95% confidence interval for the regression estimate. *** denote p < 0.001, * p < 0.05 and ns not significant.

To assess whether the acoustic similarity of the calls could explain the difference in the observed spiking activity in HVC neurons, we projected the playbacks into PCA space and used the silhouette score to determine the optimal number of clusters, which was n = 2. We then applied a k-means clustering to identify these two clusters. The results showed no systematic separation of calls based on their familiarity (familiar, unfamiliar, or both, S3 Fig). This indicates that HVC responses to calls seem to be driven more by social context and familiarity than by a categorical difference in the calls' acoustic properties. To explore the temporal relationship between the playback-evoked activity observed in the different neuron types, we cross-correlated the activity of interneurons with the one of projection neurons in individual birds. We found that more than half of the interneurons with auditory-evoked activity (91/169, Fig 2) were most strongly correlated with projection neurons at negative time lags (S10 Fig), indicating that interneuron activity often preceded projection neuron responses. Moreover, correlations were stronger when interneurons led projection neurons, indicating that interneurons provide temporally precise inhibitory input that can shape projection neuron activity.

Our results demonstrate that while both interneurons and projection neurons in HVC exhibit auditory-evoked responses to conspecific calls, interneurons exhibit a stronger and more nuanced sensitivity to socially salient vocalizations. Furthermore, classification analyses revealed that interneuron activity patterns contain sufficient information to decode caller identity with an accuracy above chance, a distinction not observed in projection neurons. These findings are consistent with a mediating role of interneurons in the integration of caller-specific auditory information within the vocal premotor nucleus HVC.

### Interneuron activity and behavioral responses to call playbacks are correlated

Given the distinct modulations observed in interneuron activity and in behavioral responses to familiar and unfamiliar call playbacks, we next explored the relationship between neural and vocal responses. By conducting corresponding behavioral and electrophysiological experiments in the same animals, we were able to compare call response characteristics to the neural dynamics elicited by the same stimuli during subsequent exposure. We correlated the behavioral features used to quantify vocal responses (response probability, response latency and latency variability, Fig 1d, 1e and 1f), and the interneuron activity parameters that were modulated by familiarity (mean firing rate, maximum firing rate and response duration, Fig 2f). We found strong correspondence between the activity of HVC interneurons in response to different calls and the observed differences in vocal responses to those calls. Specifically, behavioral call response probability was positively correlated with all neural features examined (Fig 4 top row). In contrast, call latency and variability showed a negative correlation with the features of interneuron activity (Fig 4, middle and bottom row). In other words, increased frequency, speed, and consistency of vocal responses to familiar call playbacks were associated with higher mean and peak firing rates, as well as prolonged activity in HVC interneurons. Together, these results suggest that the patterns of HVC interneuron activity that represent caller familiarity during listening also vary in relation to context-dependent call production.

## Discussion

We examined the neural circuit dynamics underlying caller-specific vocal responses. Our work builds on previous research showing that zebra finches prioritize communication with socially relevant individuals [11,12,14,17]. We

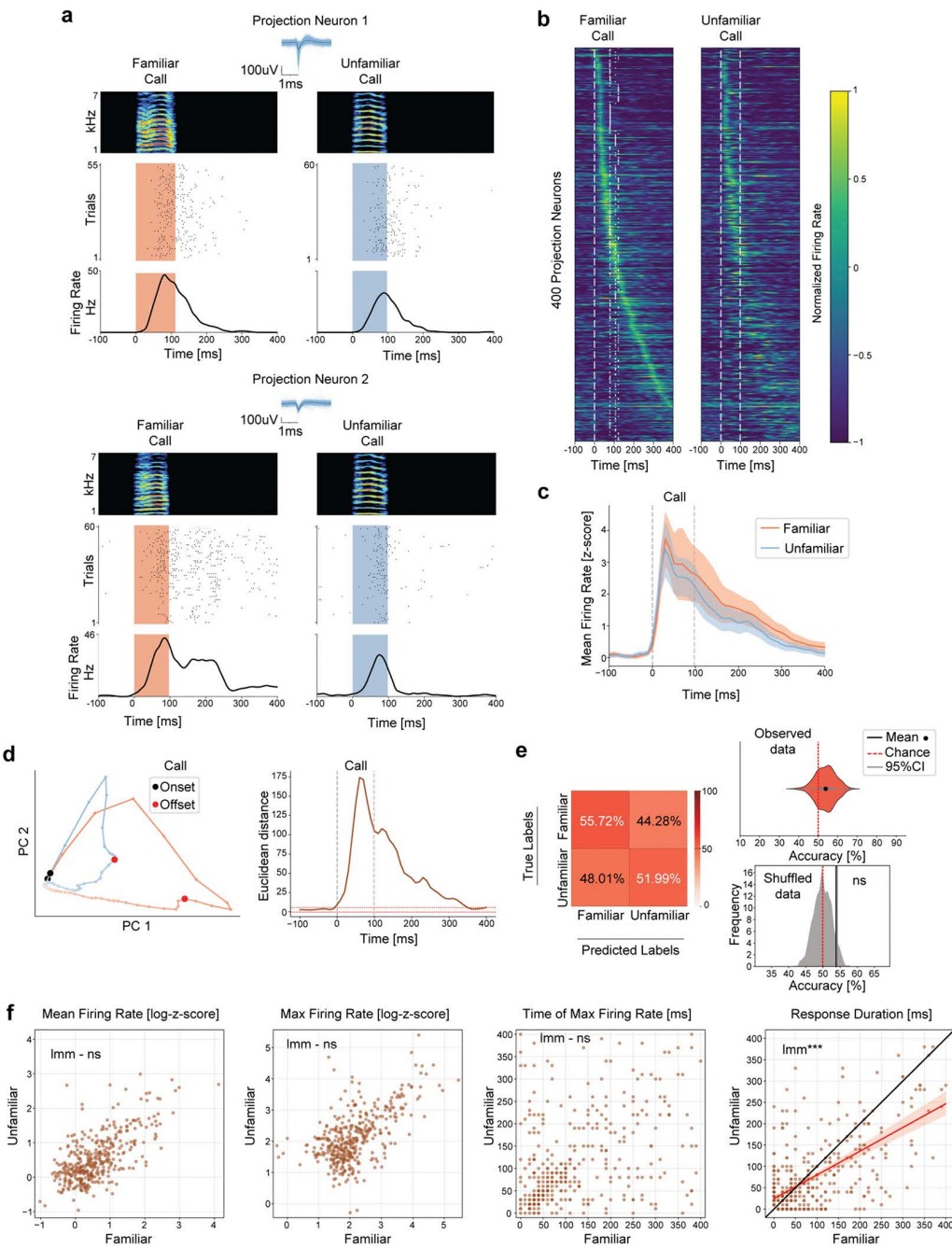

**Fig 3. Projection neuron activity differs based on caller familiarity but encodes less information about the social context (compared to HVC interneuron activity). (a)** Example cells recorded during familiar and unfamiliar call playbacks. Top: Spike-sorted waveforms with average (black) and spectrogram of call playback used. Middle: Spike dot raster plot. Bottom: peri-stimulus time histogram (PSTH). **(b)** Normalized firing rate for projection neurons that changed activity beyond two standard deviations from baseline (400/555 Projection neurons, recordings = 9, birds = 8). Neurons are ordered by their peak firing time during familiar call playbacks. The same neuron order is maintained for unfamiliar call playback, showing corresponding activity patterns across conditions. White dashed lines depict call onsets and offsets. **(c)** Average normalized firing rate (z-score) across significantly responsive neurons. Shaded area represents the 95% confidence intervals (1.96*standard error). **(d)** Left: Response trajectories in PC space for neurons shown in b (Variance explained by first 2PCs = 56.96%). Lines connecting dots represent 10ms. Right: Euclidean distance between conditions in PC space across time. Red dotted lines represent ±2std from mean baseline values. **(e)** Average classification accuracy for call playback familiarity based on the firing rate of neurons shown in b (time window = 0 to 400ms from playback onset, model = support vector machine, iterations = 1000, test size = 0.1). Left: Confusion matrix. Top Right: Distribution of accuracies across runs (53.85 ± 5.07%). Bottom Right: Kernel density estimate distribution derived from shuffled data.

The solid gray line indicates the 95% confidence interval of the shuffled distribution (54.13%), while the black solid line represents the mean accuracy of the observed data (53.85%, permutation test, p = 0.069). Chance level = 49.9%. **(f)** Distribution of different features extracted from the neural activity. Conditions compared using linear mixed model (lmm, see S8 Fig). Firing rate (familiar) = 1.54 ± 3.94 (z score), firing rate (unfamiliar) = 1.24 ± 2.46 (z score), lmm, p = 0.059. Max firing rate (familiar) = 12.20 ± 19.62 (z score), max firing rate (unfamiliar) = 11.51 ± 17.79 (z score), lmm, p = 0.44. Time of max firing rate (familiar) = 113.12 ± 104.68ms, time of max firing rate (unfamiliar) = 104.45 ± 97.98ms, lmm, p = 0.11. Response duration (familiar) = 88.22 ± 95.33ms, response duration (unfamiliar) = 73.32 ± 82.97ms, lmm, p = 0. Black line depicts the identity line where slope = 1. Red line represents fitted regression line for significant comparisons and shaded region shows the 95% confidence interval for the regression estimate. *** denote p < 0.001, and ns not significant.

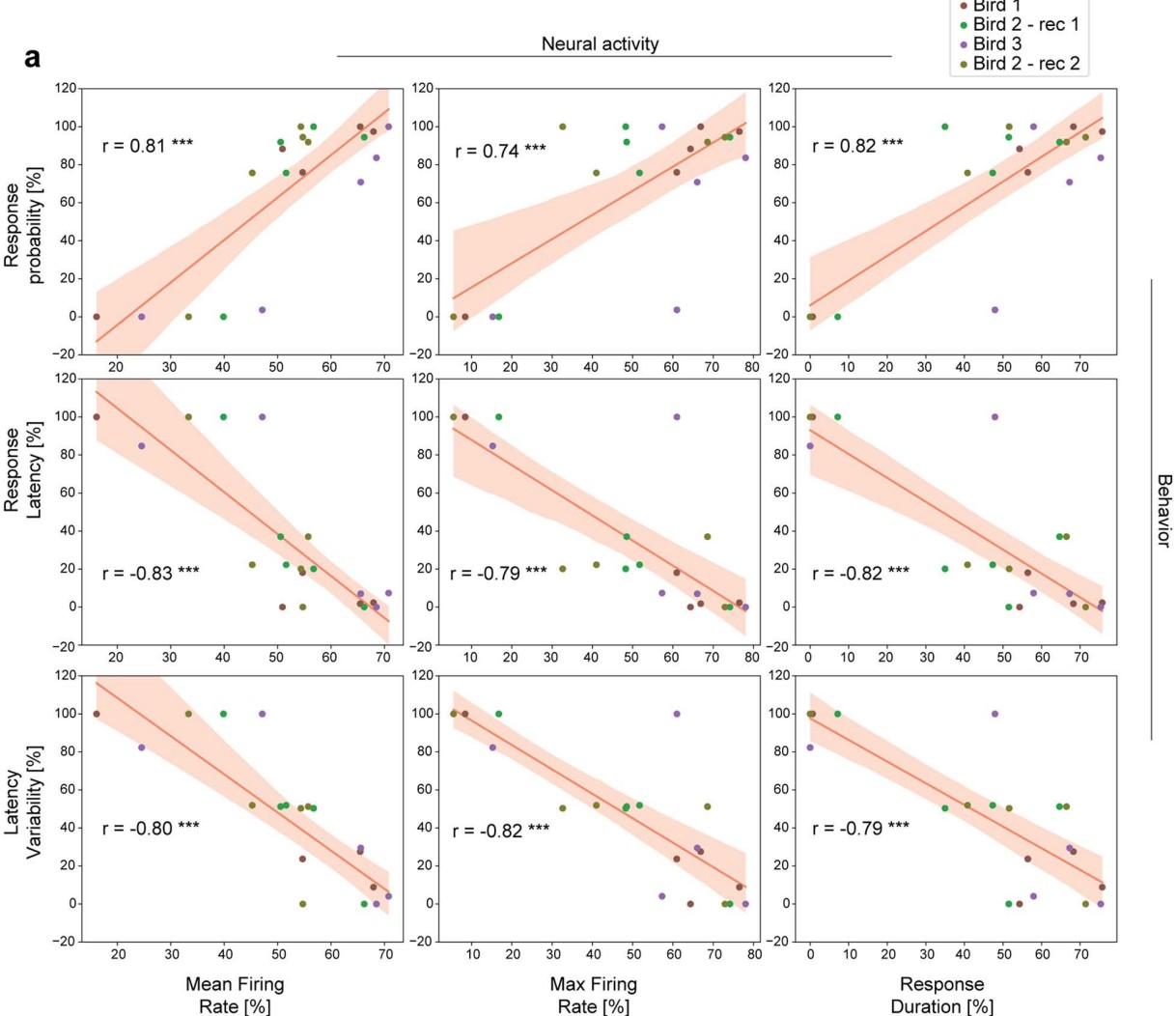

**Fig 4. Variations in call responses correlate with interneuron activity changes across different playbacks. (a)** Correlation between normalized features used to describe vocal responses (behavior) and neural activity (n = 4 sessions, across 3 birds). Behavioral data was normalized by bird, while neural recordings were normalized for each neuron individually. Neural parameters plotted represent the mean values obtained for all recorded interneurons (n = 69). Kolmogorov-Smirnov test used to test for normality, then Pearson correlation coefficient calculated (shown inside cells for each correlation, see S11 Fig). Bonferroni correction used to control for multiple comparisons. Corrected p values denote *** p < 0.001.

experimentally induced similar vocal biases, observing that birds responded more quickly and frequently to calls from co-housed birds compared to unfamiliar ones. After establishing this context-dependent behavior (Fig 1), we recorded the activity of individual neurons in the forebrain nucleus HVC, and found that firing patterns of individual neurons are differentially modulated by the familiarity of the calls (Figs 2 and 3). This demonstrates a functional link between social context and the neural circuit activity that guides vocal behavior (Fig 4).

Classically, HVC has been studied for its role in the production of learned courtship song [32]. However, HVC has also been implicated in regulating the timing of innate social calls during vocal interactions [8,21,28]. HVC is known to receive auditory inputs that are relevant for vocal behavior, specifically during development [33–37]. Our electrophysiological recordings provide further evidence that a large proportion of HVC neurons receive auditory information in awake adult birds, with more than 70% of recorded cells (putative interneurons and projection neurons) responding to calls (Figs 2 and 3). This is consistent with recent studies reporting call-evoked responses in HVC of awake birds [21,28,29], and highlights HVC as a site of continuous auditory-motor integration, capable of representing information about both heard and produced vocalizations even after the critical period for song learning.

The current study further demonstrates that auditory-related HVC activity can be context-dependent and sensitive to recent social experience. Notably, we found that interneuron activity is differentially modulated by call playbacks, depending on the familiarity of the caller, with familiar signals eliciting stronger and longer responses (Fig 2). Furthermore, we found that elevated interneuron activity can persist for hundreds of milliseconds after playback offsets. Variations in this sustained activity may be indicative of the differences in the social salience of heard calls for each bird, rather than solely representing the transient low-level acoustic properties of the stimuli (exemplified by differences in behavioral responses to identical stimuli in S12 and S3 Figs). Persistent interneuron activity occurs on behaviorally relevant timescales, largely coinciding with the typical window of call response latencies (200–450ms) [19,20], as well as with HVC neuron activity that precedes call production [8]. Given that inhibitory regulation of this premotor activity has been associated with call timing and auditory suppression of calling [21], the observed differences in call-evoked interneuron responses provide a plausible mechanism for call-specific control of vocal replies in the presence of multiple callers. Since birds were head-fixed during our neural recordings and did not actively engage in vocal turn taking, we speculate that the increased persistence in interneuron activity might reflect a stronger activation of the withholding mechanism, accompanying and counteracting the elevated vocal response drive for familiar calls. We hypothesize that the majority of call-responsive HVC interneurons are likely to be parvalbumin-expressing (PV) GABAergic neurons, as they have been shown to be modulated during song playback and have been implicated in gating the impact of sensory inputs onto other HVC neurons [36,38]. Conversely, call-responsive interneurons are less likely to be somatostatin-expressing (SST) interneurons. Although SST interneurons in HVC can strongly reflect arousal states, such states would presumably have similar effects across call stimuli, given their pseudorandom presentation and relatively short inter-call intervals (1s).

Projection neurons showed less prominent differential tuning to call familiarity (Fig 3), perhaps reflecting that these neurons, which primarily increase their firing during vocal production, exhibit less auditory-driven activity when a bird is listening but not vocally responding to calls (as in our head-fixed recordings). Correspondingly, the HVC projection neurons recorded are likely those projecting to Area X ($HVC_x$) given their spontaneous firing rate (1.93 ± 3.8 Hz), and considering that HVC neurons projecting to nucleus RA ($HVC_{RA}$) are not active in awake conditions, except for when the bird vocalizes [32]. Our cross-correlation analysis (S10 Fig) revealed that playback evoked responses of HVC interneurons tend to precede the responses of projection neuronss. Since HVC interneurons and projection neurons are tightly interconnected [39], this may reflect a motif of feed-forward inhibition from PV neurons [21] that receive context-specific auditory inputs. The question remains whether the social modulation of neuron activity elicited by conspecific calls is a property arising in HVC, or if it is inherited from upstream auditory areas. Previous work has demonstrated that neurons in the caudomedial nidopallium (NCM), a higher-order auditory region indirectly upstream from HVC, respond differently to familiar versus less familiar female distance calls, but only in the presence of an audience [40]. It is therefore likely that differences in auditory

responses would also be present in signals arriving in HVC via intermediate nucleus NIf. Our findings would then indicate that these inputs may be stronger for HVC interneurons than projection neurons, thereby introducing a means of context-dependent behavioral control at the level of HVC.

Lastly, we found strong correlations between HVC interneuron activity and vocal responses triggered by the same call playbacks during behavioral experiments, for a given bird (Fig 4), suggesting the specificity of interneuron responses observed while a bird is listening to calls might underlie the biases found in vocal behavior. Together, these findings further expand HVC's role as an auditory-motor region that integrates context-relevant auditory information to flexibly guide vocal behavior during social interactions. Future work is needed to determine how auditory information affects interneuron and projection neuron activity in HVC, along with call production during dynamic vocal exchanges.

The ability to utilize social auditory information to rapidly affect motor behavior in a context-dependent manner can serve ecological functions related to kin recognition [41,42], territorial defense [43], vocal labeling [44], and pair bonding [18]. Various studies have shown that other species also extract contextual cues from acoustic signals, relating to individual identity [45–48], sex [49–51] and group membership [52,53] to guide communication signals. Our study contributes to the understanding of how innate behaviors like calling can be modulated by higher order auditory input to enable flexible social communication.

## Materials and methods

### Ethics statement

All animal care and procedures were performed with the ethical approval of the Max Planck Institute for Biological Intelligence and the Regierungspräsidium Oberbayern (ROB-55.2-2532.Vet_02-18-182 and ROB-55.2-2532.Vet_02-21-201).

### Animals

Animals were housed in a controlled environment with a 14/10-hour light/dark cycle and were provided with food, water, and grit *ad libitum.* Birds were acquired from the breeding facility at the Max Planck Institute for Biological Intelligence - Seewiesen. Birds used in the behavioral and electrophysiological experiments were raised by their biological parents in a mixed-sex aviary. Then, taken as adults with at least 100 days post hatch. The adult male birds used in the current study were housed with female adult birds for the duration of the experiments.

### Playbacks

All playbacks used for both behavioral and extracellular recording experiments were presented at 55–65 dB through a speaker placed close to the experimental bird. Playbacks used were recorded from the sound box were an adult male and female were kept, and they were taken from stack call interactions between them. The vocalization sounds obtained were then normalized, band pass filtered (300Hz-14kHz), and overlaid with a 20 kHz pure tone pulse (beyond zebra finch auditory range and used to facilitate automated detection and analysis of behavioral data) [54,55].

### Behavioral experiment

Individual pairs of opposite sex birds (7 males and 6 females used in total) were housed (60×60×120 cm sound-attenuated box) together for a minimum of 5 days to encourage coordinated calling and responsivity to the familiar birds' vocalizations [18]. Then, a divider with a mirror was placed in between the 2-compartment sound box. This separated the pair visually, but kept acoustic communication intact. For the next 4 days, playback sessions were performed to quantify call responsivity of the male bird. The speaker was placed outside of the box close to the female compartment. Before the start of each session, the female was temporarily removed from the sound box and 20 blocks of playbacks were presented. Each block consisted of 6 repetitions of each playback sound in a pseudorandom order. A 60 second period

of silence was introduced between blocks and 1s between individual playbacks. The call playbacks used were taken from the stack call of the familiar female and stack calls from other unfamiliar birds (male and female). We designed the experiment so that each female call playback was familiar for a given bird and unfamiliar to the others tested (exemplified in S12 Fig). Each bird was presented in total with 5 different playbacks from a collection of 9 sounds that included a familiar call, 2 unfamiliar male calls, 5 unfamiliar female calls, and a catch playback (silent period). During the session, playbacks and calls produced by the male were recorded with a miniature cardioid microphone (audio-technica). At the end of each session, the female bird was brought back to the home box, which was generally followed by a vocal exchange between the pair. 3 of the 7 male birds used in the behavioral experiments were also used for extracellular recordings.

## Surgery

In preparation for head-fixed awake electrophysiological recordings, male zebra finches (n = 8 birds) were implanted with a custom-made stainless steel headplate. First, male birds were anesthetized with isoflurane (1–3% in oxygen). Then, HVC was located based on stereotactic coordinates and a craniotomy and durotomy were performed (right hemisphere, head angle: 45–55°, 0.1-0.5mm anterior, 2.25-2.4mm lateral, relative to the bifurcation of the midsagittal sinus). The location of HVC was later confirmed by histology. For signal reference, a small craniotomy above the cerebellum was made and a ground wire (0.05 mm bare, silver, Science Products) was placed between the skull and the dura mater. All craniotomies were then sealed with a silicone elastomer (Kwik-Cast, WPI) to prevent desiccation. Finally, the headplate was implanted using light acrylic and dental cement (Paladur, Kulzer International). All animals were taken back to their housing enclosure with a companion female bird for at least 24 hours post-surgery and were monitored to confirm recovery before continuing experiments.

## Electrophysiological recordings

Neuropixels 1.0 probes [56] were used to record in awake, head-fixed adult birds (n = 9 sessions, birds = 8). Animals were placed in a foam-lined container to restrict movement. With this configuration, the bird's head was fixed with 3 set screws to allow stable access to the recording target. In preparation for the recording, the previously identified location of HVC was surrounded by a well built with silicone elastomer around the craniotomy, that was then filled with phosphate buffered saline (PBS, Carl Roth). Duramater was carefully cleared using a dura-pick to gain visible access to HVC. To confirm HVC targeting, DiI (ab145311, Abcam) diluted in isopropyl alcohol was applied to the backplane of the probe before insertion. The recording was carried out using an external reference and shared ground-reference configuration. The reference-ground wire was connected to the wire on top of the cerebellum using gold pins. Two TTL pulses were used, one triggered by the playback onset using an Arduino Uno, and another 1s squared wave pulse generated by the Neuropixels chassis. A New Scale micromanipulator was used to insert the probe at an angle of 0–10°. Once it reached the desired depth, a 20 minutes window without recording was given for stabilization. Then, the familiar female and unfamiliar playbacks were presented in pseudorandom order for at least 48 trials each. Again, a 1s period of silence separated each playback. Each bird was presented with a familiar call playback, an unfamiliar one, and a control stimulus (20kHz pure tone or catch consisting of 1s period of silence, S5 and S6 Figs). For electrophysiological experiments, only animals in Fig 4 received 5 playbacks, 1 familiar female, 1 unfamiliar female, 2 unfamiliar male, and a control stimulus. The speaker was placed in front of the head-fixed bird at a distance of at least 25 cm. Audio, and neural signals were acquired simultaneously using a breakout box (NI-BNC-2110) and aligned offline. At the end of the experiment, birds were immediately perfused and the brain was extracted for histology.

## Histology

After perfusion, brains were kept in 4% paraformaldehyde (PFA, Sigma-Aldrich) and preserved at 4° until further processing. Next, brains were transferred into PBS, and 15% and 30% sucrose solutions in 24-hour steps each. Then, the brain

was split into 2 hemispheres, and only the right one (recorded side) was cut in 40 μm slices using a freezing microtome. Slices were kept in PBS, and, after rinsing in deionized water, immediately mounted on Superfrost adhesion microscope slides (Epredia), coated with fluorescence mounting medium (Dako), and secured with coverslips (Epredia). To confirm site insertion, pictures were taken at 2.5 magnification using an epifluorescence microscope (Leica). For further analysis, only recording sites that covered the ventral length of HVC were considered. To determine HVC channels we considered the depth of insertion during recordings, the probe track observed in histology (S13 Fig), and the spatial extent of spiking activity across Neuropixels channels.

## Data acquisition

**Audio recordings.** All audio data during experiments was acquired using a custom-made MATLAB script. For the behavioral experiments, call instances produced by the male were manually labeled and only trials answered with contact calls were counted. This implies that any trials were birds responded with distance calls or song were excluded.

**Electrophysiological recordings.** Electrophysiological data was acquired using SpikeGLX and spike detection and clustering was completed with Kilosort 2.5 [57].

Behavioral and neural experiments in Fig 4 were performed independently, but using the same birds. Specifically, behavioral measures were taken while animals were in a sound box as described in behavioral experiment section, and neural recordings were conducted in head-fixed conditions (thus, birds do not readily call back) according to electrophysiological recordings description above.

To avoid conflating motor and sensory activity, the subset of trials with a vocal response to the playback were excluded from analyses of neural recordings.

## Data analysis

**Audio data analysis.** Acoustic similarity between the playbacks used for the experiments was assessed using Principal Component analysis. We conducted PCA using the sound waveform time bins as features. This gave us 7 main PCs explaining 93.47% of the variance. Projecting our playbacks into this PC space, we then determined the optimal number of clusters to understand whether these calls were categorized in the same cluster (indicating acoustic similarity) or in different clusters (indicating acoustic differences). We did this using the silhouette score, which suggested two clusters (n = 2) as the optimal number. We then used a k-means clustering algorithm to identify these two clusters (S3 Fig).

**Electrophysiological data analysis.** Doubled counted spikes were removed using ecephys spike sorting repository. Resulting clusters were manually curated using phy, and single cells were obtained after considering the following quality metrics parameters: an interspike interval violation index (isi) > 0.3 [58], amplitude cutoff distribution < 0.5, and presence ratio > 70% [59]. To differentiate projection neurons from interneurons, we quantified waveform durations (trough-to-peak) using custom-made python scripts and SpikeInterface [60], and separated narrow (mean duration: 0.20ms) versus broad (mean duration: 0.52ms) spiking neurons [38,61] setting a threshold at 0.35ms (S4 Fig). In addition, we confirmed that the classified neurons matched previously described spontaneous firing rates typical for both types of neurons [62,63].

Firing rates were calculated with a bin size of 10ms and smoothed using Gaussian convolution (kernel width: 3 bins), except for Figs 2a and 3a where a 1ms bin was used and smoothing was done with a Savitzky-Golay filter (51ms window, polynomial order 1). Z-scores were calculated as: Z = (Firing Rate − Mean Baseline Firing Rate)/ Baseline Standard Deviation. Significantly responsive neurons were defined as cells showing activity deviating ±2std (z scores) from baseline for at least 1-time bin and within a window from 0ms at playback onset to 400ms.

We performed PCA using the normalized Peristimulus Time Histograms (PSTHs) of responsive neurons. Then, for visualization, trajectories were smoothed using convolution with a Gaussian kernel (width = 3). To determine Euclidean Distance significant periods, we took the baseline values (100ms before call onset) for each PC counted and shuffled

them 1000 times. With this shuffled distribution, we calculated the mean and considered 2 std deviations away from this as periods where Euclidean Distance is significant.

## Statistical tests

All statistical tests used are specified in the main text and captions. If a statistical test for normally distributed data was used, the dataset was first confirmed to be normally distributed by a Kolmogorov-Smirnov test. Unless explicitly mentioned, all measures of central tendency reported are medians. However, when standard deviation ranges are provided, means are used.

## Classification analysis

For classification tasks, the data was min-max normalized based on the responses to the different stimuli presented. Behavioral data was normalized individually for each bird, and using behavioral parameters: peak response probability, response latency, and latency variability for familiar vs. unfamiliar calls. As there were more instances of unfamiliar playbacks during behavioral experiments, we randomly subsampled them for each iteration to fit the familiar playback size. Then, we used stratification to split the data considering 50% of it as training. Data from neural recordings was normalized on a per-neuron basis using firing rate, and 90% was used for training. To get a reliable measure of the accuracy of predictions, all classifiers were run 1000 times with different train-test datasets randomly selected. To evaluate the significance of observed classifier accuracies, we generated a shuffle distribution by randomly permuting labels 1,000 times and running the classifier 10 times for each iteration. We assessed significance by calculating the proportion of shuffled accuracies exceeding the observed mean accuracy (p value). For visualization, we plotted the observed mean accuracy along with the 95% confidence interval derived from the shuffled distribution, calculated as the value below which 95% of the shuffled accuracies fall. Chance level was defined as the mean accuracy of the shuffled data.

## Correlation analysis

For correlation analysis between behavioral and neuronal data, data from behavioral and neural features was min-max normalized based on the values for the different stimuli presented and scaled from 0-100 to represent percentage values.

For cross correlation analysis of neuron activity across different neuron types, we calculated coefficients (per recording id, recordings = 9, birds = 8, interneurons = 169, projection neurons = 400) by taking each interneuron firing rate (1ms time bins) around the presented playbacks (familiar and unfamiliar, time window = -100–400ms in relation to playback onset). Cross-correlations were computed between mean-centered interneuron and projection neuron firing rates. To allow comparison across neurons, cross-correlation values were normalized by the product of the signal length and the standard deviations of both signals, yielding correlation coefficients bounded between –1 and 1. Each interneuron was compared to all corresponding (simultaneously recorded) projection neurons, and the time lag with the highest correlation was taken. Then, we obtained the mean correlation and time lag across all these interneuron-projection neuron comparisons and tested if there was any correlation between these 2 parameters (S10 Fig).

## Supporting information

**S1 Fig. Call response feature variability across experimental days** (a) Mean values for peak response probabilities to familiar and unfamiliar call playbacks by experimental day (n = 9, birds = 7). ANOVA, familiarity effect p = 0.003, day effect p = 0.001, interaction p = 0.68. (b, c) Same arrangement as in a, but for response latencies and latency variability. Response latency ANOVA, familiarity effect p = 0.002, day effect p = 0.41, interaction p = 0.49. Response variability ANOVA, familiarity effect p = 0.04, day effect p = 0.18, interaction p = 0.86. Error bars depict standard error.
(TIF)

**S2 Fig. Modulation of vocal behavior by caller familiarity divided by sex.** (a) Peak response probabilities to different call playbacks (n=9, birds=7, days=4). Response probability (familiar)=0.117, (unfamiliar male)=0.094, (unfamiliar female)=0.091. Wilcoxon signed-rank test, familiar versus unfamiliar male, p=0.003. Familiar versus unfamiliar female, p=0.25. (b, c) Response latencies and response latency variability. Response latency (familiar)=306ms, (unfamiliar male)=352ms, (unfamiliar female)=362ms. Wilcoxon signed-rank test, familiar versus unfamiliar male, p=0.019. Familiar versus unfamiliar female, p=0.039. Latency variability (familiar)=246ms, (unfamiliar male)=280ms, (unfamiliar male)=256ms. Wilcoxon signed-rank test, familiar versus unfamiliar male, p=0.019. Familiar versus unfamiliar female, p=0.097. Blue solid dots represent mean values across multiple unfamiliar playbacks when applicable, and error bars standard error (sem). * denote p<0.05, ** p<0.01, and ns non significant.
(TIF)

**S3 Fig. Playback acoustic similarity** (a) Playbacks projected into principal component space based on their acoustic waveforms (10 playbacks, variance explained by first 2 PCs=51.03%). Each point represents a single sound. Dots labeled to indicate if playback was used as a familiar, unfamiliar or both during the experiments. (b) Same arrangement as in a, but points are labeled according to cluster id. Using silhouette score, we determined the optimal number of clusters, and identified them using kmeans (see Methods). Only one playback was assigned to Cluster 1 (bottom right), while all other stimuli were labeled as Cluster 2.
(TIF)

**S4 Fig. Neuron classification based on waveform duration.** (a) Histogram of trough-to-peak duration for all HVC cells recorded (neurons: 765, recordings=9, birds=8, bin size=0.06ms). Mean interneuron waveform duration=0.20ms. Mean projection neuron waveform duration=0.52ms. (b) Top: Example putative interneuron and projection neuron. For each cell, a subset of 1000 spikes were selected randomly and used in the stack traces plotted. The solid black line represents the mean trace. Bottom: Scatterplot of waveform duration and spontaneous firing rate for cells shown in a. Each solid dot corresponds to 1 neuron. Mean firing rate for interneurons: 9.51±8.37Hz. Mean firing rate for projection neurons: 1.93±3.8Hz. Red dashed line indicates the threshold chosen to separate narrow and broad spiking neurons (interneurons=210, projection neurons=555).
(TIF)

**S5 Fig. Interneuron activity during playbacks and control stimuli.** (a) Normalized firing rate for interneurons deviating 2 standard deviations from baseline during the familiar and unfamiliar call playback (recordings=5, birds=5). Neurons are ordered by their peak firing time during familiar call playback. The same neuron order is maintained for the other stimuli, showing corresponding activity patterns across conditions. White dashed lines depict call onsets and offsets. The pure tone represents a 20kHz sound presented as a control. (b) Same arrangement as in a), but for a different set of birds (recordings=4, birds=3) presented with catch trials (trials with silent playback).
(TIF)

**S6 Fig. Projection neuron activity during playbacks and control stimuli.** (a) Normalized firing rate for projection neurons deviating 2 standard deviations from baseline during the familiar and unfamiliar call playback (recordings=5, birds=5). Neurons are ordered by their peak firing time during familiar call playback. The same neuron order is maintained for the other stimuli, showing corresponding activity patterns across conditions. White dashed lines depict call onsets and offsets. The pure tone represents a 20kHz sound presented as a control. (b) Same arrangement as in a), but for a different set of birds (recordings=4, birds=3) presented with catch trials (trials with silent playback).
(TIF)

**S7 Fig. Interneuron-based decoding of familiarity across playback and post-playback periods** (a) Average classification accuracy for call playback familiarity based on the firing rate of neurons shown in Fig 2b (time window=0–100ms

from playback onset, model = support vector machine, iterations = 1000, test size = 0.1). Left: Confusion matrix. Top Right: Distribution of accuracies across runs (57.36 ± 7.54%). Bottom Right: Kernel density estimate distribution derived from shuffled data. The solid gray line indicates the 95% confidence interval of the shuffled distribution (56.47%), while the black solid line represents the mean accuracy of the observed data (57.36%, permutation test, p = 0.03). When including only during-playback responses (0–100ms), the classifier predicted which call was presented with an accuracy significantly higher than chance. Chance level = 50%. (b) Same as in a, but for the time window = 100ms to 400ms, corresponding to the post-playback period. Top Right: Distribution of accuracies across runs (55.32 ± 7.94%). Bottom Right: 95% confidence interval of the shuffled distribution (56.76%). Mean accuracy of the observed data (55.32%, permutation test, p = 0.085). When including only the late responses (100–400ms), the classifier was unable to reliably predict which call was presented. Chance level = 50%. * denotes p < 0.05 from permutation test, while ns non-significant.
(TIF)

**S8 Fig. Summary statistics of linear mixed models.**
(TIF)

**S9 Fig. Neural features classify accurately calls based on familiarity.** Average classification accuracy to classify call playback familiarity based on neural features that differed for familiar and unfamiliar playbacks (See Fig 2f & 3f) Model = support vector machine, iterations = 1000, test size = 0.1. (a) Familiarity classification using interneurons and based on neural features mean firing rate, max firing rate, and response duration. Left: Confusion matrix. Top Right: Distribution of accuracies across runs (62.76 ± 7.9%). Bottom Right: Kernel density estimate distribution derived from shuffled data. The solid gray line indicates the 95% confidence interval of the shuffled distribution (56.47%), while the black solid line represents the mean accuracy of the observed data (62.76%, permutation test, p = 0.002). Chance level = 49.76%. (b) Same arrangement as in a), but for projection neurons and with a classifier based on neural features mean firing rate and response duration. Observed data mean = 58.01 ± 5.22, 95%CI of shuffled data = 54.51%, permutation test, p = 0.002. Chance level = 50%.
(TIF)

**S10 Fig. Cross correlation of interneuron and projection neuron activity across different time lags.** (a) Scatterplot of mean correlation coefficients and mean time lags for all interneurons with auditory-evoked activity recorded (recordings = 8, birds = 9, interneurons = 169, projection neurons = 400). Each dot represents an interneuron and how it correlates on average with all other simultaneously recorded projection neurons and at what time lag. Red line represents fitted regression line and shaded region shows the 95% confidence interval for the regression estimate. *** denote p < 0.001 from Spearman correlation analysis (correlation coefficient = -0.34, p = 7.2 × 10$^{-6}$).
(TIF)

**S11 Fig. Summary statistics of correlations between behavioral and neural parameters.** Kolmogorov-Smirnov test used to test for normality, then Pearson correlation coefficient calculated. Bonferroni correction used to control for multiple comparisons.
(TIF)

**S12 Fig. Birds preferentially respond to familiar playbacks.** Response profiles from two example birds presented with a familiar and unfamiliar playback. Playback identity is color coded. (a) Top panel: Call responses to call playbacks presented once per second (exemplified by data from the initial two days). Bottom panel: Call response probability (across four days). (b) Same arrangement as in a, but for a different experimental bird.
(TIF)

**S13 Fig. Brain histology.** Example trace from a neural recording in HVC. Only channel sites within HVC were considered for analysis (see Methods).
(TIF)

## Acknowledgments

We thank all Vallentin lab members for comments during the development of the project, and Lidia Lopez for designing the illustrations used in this manuscript.

## Author contributions

**Conceptualization:** Carlos M. Gomez-Guzman, Daniela Vallentin, Jonathan I. Benichov.

**Data curation:** Carlos M. Gomez-Guzman, Daniela Vallentin, Jonathan I. Benichov.

**Formal analysis:** Carlos M. Gomez-Guzman, Daniela Vallentin, Jonathan I. Benichov.

**Funding acquisition:** Daniela Vallentin, Jonathan I. Benichov.

**Investigation:** Carlos M. Gomez-Guzman, Jonathan I. Benichov.

**Methodology:** Carlos M. Gomez-Guzman, Jonathan I. Benichov.

**Project administration:** Daniela Vallentin, Jonathan I. Benichov.

**Resources:** Daniela Vallentin, Jonathan I. Benichov.

**Supervision:** Daniela Vallentin, Jonathan I. Benichov.

**Validation:** Carlos M. Gomez-Guzman, Jonathan I. Benichov.

**Visualization:** Carlos M. Gomez-Guzman, Daniela Vallentin, Jonathan I. Benichov.

**Writing – original draft:** Carlos M. Gomez-Guzman, Daniela Vallentin, Jonathan I. Benichov.

**Writing – review & editing:** Carlos M. Gomez-Guzman, Daniela Vallentin, Jonathan I. Benichov.

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
