## [Decision Letter · Decision Letter 0]

29 Jul 2025

PCOMPBIOL-D-25-00720

Social familiarity strengthens neural and vocal responses to conspecific calls in zebra finches

PLOS Computational Biology

Dear Dr. Gomez-Guzman, Dr. Vallentin, and Dr. Benichov,

Thank you for submitting your manuscript to PLOS Computational Biology. After careful consideration, we feel that it has merit but does not fully meet PLOS Computational Biology's publication criteria as it currently stands. Therefore, we invite you to submit a revised version of the manuscript that addresses the points raised during the review process.

Please submit your revised manuscript within 60 days (by the end of August 2025). If you will need more time than this to complete your revisions, please reply to this message or contact the journal office at ploscompbiol@plos.org. Please include the following items when submitting your revised manuscript:

We look forward to receiving your revised manuscript.

Kind regards,

Iris Vilares

Academic Editor

PLOS Computational Biology

Joseph Ayers

Section Editor

PLOS Computational Biology

**Journal Requirements:**

4) We notice that your supplementary Figures are included in the manuscript file. Please remove them and upload them with the file type 'Supporting Information'. Please ensure that each Supporting Information file has a legend listed in the manuscript after the references list.

Potential Copyright Issues:

i) Please confirm (a) that you are the photographer of S2, or (b) provide written permission from the photographer to publish the photo(s) under our CC BY 4.0 license.

ii) Figures 1a, 1b, 1c, and S1. Please confirm whether you drew the images / clip-art within the figure panels by hand. If you did not draw the images, please provide (a) a link to the source of the images or icons and their license / terms of use; or (b) written permission from the copyright holder to publish the images or icons under our CC BY 4.0 license. Alternatively, you may replace the images with open source alternatives. See these open source resources you may use to replace images / clip-art:

2) If any authors received a salary from any of your funders, please state which authors and which funders.

7) Regarding your Competing Interests statement, thank you for indicating "Nothing to disclose." If you have no competing interests to declare, please state "The authors have declared that no competing interests exist".

8) Please ensure that the funders and grant numbers match between the Financial Disclosure field and the Funding Information tab in your submission form. Note that the funders must be provided in the same order in both places as well. Currently ," Emmy Noether grant (Project number VA742/2-1)" is missing from the Funding Information tab.

**Reviewers' comments:**

Reviewer's Responses to Questions

Reviewer #1: This manuscript makes an important contribution to our understanding of how social familiarity modulates vocal communication circuits in songbirds. The authors have elegantly combined behavioral paradigms with cutting-edge neural recording techniques to reveal how the HVC encodes socially relevant information that guides context-dependent vocal responses.

The authors find that call familiarity leads to more and faster behavioral responses. Correlated with these behavioral findings, HVC interneurons (and to a lesser extent, projection neurons) respond more strongly and persistently, providing a potential neural correlate for the behavioral effects.

This work represents a significant advance in the field of communication neuroscience and merits publication in PLOS Computational Biology.

I have only one minor suggestion for improvement:

The increase in response persistence for familiar calls is particularly interesting. Could the authors speculate on potential mechanisms underlying this increased persistence in the discussion section?

Reviewer #2: A critical feature of social brains is the ability to process social and contextual information and use it to guide appropriate behavioral responses. Understanding how the brain tackles these types of tasks and determines these behavioral decisions remains poorly understood. In this paper, Gomez-Guzman et al. use a clever and simple behavioral paradigm, measuring call responses to call playback, of both familiar and unfamiliar individuals, to probe how population neural responses in a well characterized audio-vocal brain structure code both for perceived call type as well as call initiation and production. They provide convincing behavioral findings that call responses to playback (e.g. latency, probability of response) can be distinguished between the two classes of stimuli (familiar vs. unfamiliar calls). These data are then followed by neuropixel recordings in nucleus HVC showing that call type (familiar vs unfamiliar) can be decoded by the population activity recorded in HVC and that these response properties can then in turn be used to predict when and if the bird will respond with a call. These findings show in this key vocal control area that the brain can very rapidly decode communication signals based on social familiarity and use this information to initiate context appropriate behavioral responses. This is a well written manuscript with generally well-designed experiments.

Primary concern:

My only major concern is for the analyses the authors perform in figure 5 to investigate whether HVC auditory evoked responses can be decoded to predict call response probability and latency. producing a call. My concern lies in the potential overlap between the auditory evoked response and premotor activity associated with the produced call. The authors should provide evidence that there is no overlap between sensory and motor activity and some form of validation that these are clearly dissociated in time. If there is overlap, then the authors should at least provide evidence that decoding of the auditory response is not biased by underlying pre-motor preparatory activity in HVC, since this could bias their findings. They should also include a raster plot where activity is aligned to the onset of call playback and a parallel raster where activity is aligned to onset of the produced call.

Other comments:

1. The authors do a nice job determining neural trajectory in their head fixed recordings in response to call presentations. It is unclear, why they do not perform the same types of analyses in figure 5 for their call back paradigm, instead relying only on features such as neural response duration and maximum and mean firing rate for their analyses? For these experiments, it is also not clear whether these animals are still head fixed and calling or tethered. This should be made clearer in the results as well as the methods section.

2. Normalized firing patterns shown for their neuronal population ranked by response onset show that about 2/3 of their neurons (interneurons as well as projection neurons) have short response latencies to both familiar and unfamiliar calls. The remaining third of the neurons have much longer onset response latencies for familiar calls but weak to no responses for unfamiliar calls. Is it possible that the classifier relies on the weight of this last set of neurons to discriminate between call types? Certainly, by eye, these long latency neurons show a striking difference in responsiveness to the different call classes.

3. Unless I missed it, the authors should provide data, maybe in the form of a UMAP plot, to show that familiar and unfamiliar calls have overlapping acoustic features and that birds (and HVC) are not simply discriminating between these two classes of calls by acoustics features but because of familiarity. This will be an important control to provide.

Reviewer #3: In this study, the authors investigate the neuronal correlates of call processing in a songbird species, the zebra finch. They first provide behavioural evidence that males zebra finches respond differently to calls emitted by familiar compared to unfamiliar conspecifics. Building on this, they then turn to large-scale electrophysiological recordings using Neuropixels probes. They focus their recordings on the HVC, a premotor area known for its critical role in the birdsong behaviour, but also involved in vocal turn-taking of zebra finches, as shown by a previous work from the same team. Here, electrophysiological recordings were made in awake head-fixed birds while they were exposed to a set of 5 distinct auditory stimuli: a call from a familiar female, a call from an unfamiliar male, a call from an unfamiliar female, a 20 kHz pure tone pulse and a silent playback. Based on the shapes of the isolated units, the authors were able to separate putative interneurons (INs) from putative projection neurons (PNs). Both INs and PNs had their spiking activity modulated by the familiarity of the calls. PNs exhibit lower level of responses compared to INs. This study thus shows that auditory stimuli conveying specific social information impact the neuronal processing within a premotor area.

Overall, I found the paper well-written and easy to follow. Results are clear and conclusions are in line with the data. Below are some comments/questions (not sorted by importance).

I wonder whether the authors could be able to provide some assessment of how familiar the familiar calls are, and how to ensure that the unfamiliar calls remain unfamiliar over the course of the study. Did you monitor the call interactions during the first five days of co-housing? If yes, on average, what is the number of call interaction per day or during the five days? In fig 1d, I guess the results are a summary of the 4 test days. Did you notice a day effect in the behavioural responses? This might be critical for the unfamiliar calls since they may become familiar after many exposures.

How distinct are the calls? It could be interesting to provide a measure of (dis)similarity between the different calls.

About the number of playbacks for each sound, it is written in the method section (“Behavioral experiment” section) that there are 20 blocks of 6 repetitions of each playback sound, during 4 days. So if I get it correctly, each sound has been played back 480 times (20x6x4). Why are there “only” 234 trials for Familiar calls, 232 for unfam call 1, 228 for unfam call 2 shown on fig 1c? Is it that only trials with a call response are shown?

In fig1d-f, the large error bars for behavioural responses to unfamiliar calls may suggest a difference between the response to the unfamiliar males vs unfamiliar females. It could be interesting to assess whether behavioural responses do differ between unfamiliar males/females. One suggestion is to show in a supplementary fig the same plot than in Fig1d-f separating unfamiliar females and males, and a statistical comparison between the two.

Did you ever have responses with song production? If yes, was it sufficiently frequent to assess whether the amount of songs produced differ according to the call familiarity and/or sex of the caller?

Line 92-95: I do not understand why the accuracy level indicates that the inherent features of the acoustic signals are not so important in predicting the individual response. Could you please provide some more info?

To provide support for the assumption made in the discussion on the modulation of PNs via a feed-forward inhibition from INs (lines 318-321), it would be interesting to take advantage of the simultaneous recordings of many units with the Neuropixels probes to compute some cross-correlation between PNs and INs, at different time lags regarding the call onset.

It could be interesting to add a small paragraph in the discussion about whether the social modulation of neuronal activity elicited by the conspecific calls (familiar/unfamiliar) is an arising property of the HVC, or if it is inherited for upstream (auditory) areas. I’m thinking of NCM whose neurons do not discriminate familiar vs unfamiliar calls, except if there is an audience (Menardy et al, Eur. J Neurosci 2014).

Line 374-375: what are the 5 different playbacks/bird should be provided here (1 familiar female, 1 unfam male, 1 unfam fem, a 20 kHz pure tone pulse and a silent playback)

Line 359: why vocalizations were overlaid with a 20 kHz pure tone pulse?

Fig 2 & 3: it would be nice to see the waveforms of the exemplar units.

Minor comment :

Fig 1g : there are no “top right” and “bottom right” panels (but middle, and right).

**Have the authors made all data and (if applicable) computational code underlying the findings in their manuscript fully available?**

Reviewer #1: Yes

Reviewer #2: Yes

Reviewer #3: **No:** Data are fully available on Github, but not the code used for analyses

PLOS authors have the option to publish the peer review history of their article (what does this mean? ). If published, this will include your full peer review and any attached files.

**Do you want your identity to be public for this peer review?** For information about this choice, including consent withdrawal, please see our Privacy Policy .

Reviewer #1: No

Reviewer #2: No

Reviewer #3: **Yes:** Nicolas Giret

**Figure resubmission:**
---

## [Decision Letter · Decision Letter 1]

21 Nov 2025

PCOMPBIOL-D-25-00720R1

Social familiarity strengthens neural and vocal responses to conspecific calls in zebra finches

PLOS Computational Biology

Dear Dr. Vallentin,

Thank you for submitting your manuscript to PLOS Computational Biology. Overall, the reviewers seem happy with the revised manuscript, and just suggested a couple of minor revisions. Therefore, we invite you to submit a revised version of the manuscript that addresses the points raised during the review process. Also, make sure that the data/code for the manuscript are available, as per PLoS policy.

Please submit your revised manuscript within 30 days (by mid November). If you will need more time than this to complete your revisions, please reply to this message or contact the journal office at ploscompbiol@plos.org. Please include the following items when submitting your revised manuscript:

We look forward to receiving your revised manuscript.

Kind regards,

Iris Vilares

Academic Editor

PLOS Computational Biology

Joseph Ayers

Section Editor

PLOS Computational Biology

**Additional Editor Comments:**

The reviewers are on average happy with the revised paper, and just suggested a couple of minor edits/revisions. Once you have it ready and address these minor edits, please submit it back - it will likely not need to go for review again. Also, please make sure that the data/code is available, as per PLoS policy.

**Reviewers' comments:**

Reviewer's Responses to Questions

**Comments to the Authors:**

Reviewer #1: The authors have done a wonderful job in addressing my concerns. I support the publication of the manuscript in PLOS Computational Biology.

Reviewer #2: I thank the authors for their careful reworking of the text to address the reviewers's comments. I also appreciated extra analyses they provided to investigate potential confounds.

I am happy with the way they addressed my comments and am now happy with this manuscript.

My two very minor comments are:

(1) It might be worth providing a bit more explanation in the legend of Supplemental Figure 3b as it took me a while noticing there we two clusters, because the dot in the bottom right is really easy to miss.

(2) I would have like to see a bit more hand holding for supplemental Figure 7. The point, for me at least, is not entirely obvious that 0-100 sec is significant whereas analysis of 100-400 sec is not. This could be made more explicit in the figure perhaps as well as in the description of the figure legend.

Reviewer #3: I would like to thank the authors for their revised version of the MS. They have properly addressed all my comments.

I have a few minor comments:

- on fig 2f, since the correlation between unfam/fam max FR is not significant (p=0.051), there should be no red and black lines.

- please add the unit of the firing rate in the legend of figs 2 & 3 (lines 188-189 & 216-218)

**Have the authors made all data and (if applicable) computational code underlying the findings in their manuscript fully available?**

Reviewer #1: None

Reviewer #2: Yes

Reviewer #3: **No:** I was unable to locate the code and data described in the MS.

PLOS authors have the option to publish the peer review history of their article (what does this mean? ). If published, this will include your full peer review and any attached files.

**Do you want your identity to be public for this peer review?** For information about this choice, including consent withdrawal, please see our Privacy Policy .

Reviewer #1: No

Reviewer #2: No

Reviewer #3: **Yes:** Nicolas Giret

**Figure resubmission:**
---

## [Editor Report · Decision Letter 2]

15 Feb 2026

Dear Dr. Vallentin,

We are pleased to inform you that your manuscript 'Social familiarity strengthens neural and vocal responses to conspecific calls in zebra finches' has been provisionally accepted for publication in PLOS Computational Biology.

Best regards,

Iris Vilares

Academic Editor

PLOS Computational Biology

Lyle Graham

Section Editor

PLOS Computational Biology

---

## [Editor Report · Acceptance letter]

PCOMPBIOL-D-25-00720R2

Social familiarity strengthens neural and vocal responses to conspecific calls in zebra finches

Dear Dr Vallentin,

I am pleased to inform you that your manuscript has been formally accepted for publication in PLOS Computational Biology. Your manuscript is now with our production department and you will be notified of the publication date in due course.

With kind regards,

Anita Estes
